# Knowledge, facilitators and barriers to cervical cancer screening among women in Uganda: a qualitative study

Rawlance Ndejjo,[1] Trasias Mukama,[1] Juliet Kiguli,[2] David Musoke[1]

[1]Department of Disease Control and Environmental Health, School of Public Health, College of Health Sciences, Makerere University, Kampala, Uganda
[2]Department of Community Health and Behavioural Sciences, School of Public Health, College of Health Sciences, Makerere University, Kampala, Uganda

**Correspondence to**
Mr. Rawlance Ndejjo;
rndejjo@musph.ac.ug

## ABSTRACT

**Objectives** To explore community knowledge, facilitators and barriers to cervical cancer screening among women in rural Uganda so as to generate data to inform interventions.

**Design** A qualitative study using focus group discussions and key informant interviews.

**Setting** Discussions and interviews carried out in the community within two districts in Eastern Uganda.

**Participants** Ten (10) focus group discussions with 119 screening-eligible women aged between 25 and 49 years and 11 key informant interviews with healthcare providers and administrators.

**Results** Study participants' knowledge about cervical cancer causes, signs and symptoms, testing methods and prevention was poor. Many participants attributed the cause of cervical cancer to use of contraception while key informants said that some believed it was due to witchcraft. Perceptions towards cervical cancer and screening were majorly positive with many participants stating that they were at risk of getting cervical cancer. The facilitators to accessing cervical cancer screening were: experiencing signs and symptoms of cervical cancer, family history of the disease and awareness of the disease/screening service. Lack of knowledge about cervical cancer and screening, health system challenges, fear of test outcome and consequences and financial constraints were barriers to cervical cancer screening.

**Conclusion** Whereas perceptions towards cervical cancer and screening were positive, knowledge of study participants on cervical cancer was poor. To improve cervical cancer screening, effort should be focused on reducing identified barriers and enhancing facilitators.

## INTRODUCTION

Globally, in 2012, cervical cancer was responsible for 265 700 deaths and 527 600 diagnoses, 85% of which occurred in developing countries.[1 2] In East Africa, it is the leading cause of cancer deaths and has the highest age-standardised incidence rates of 42.7 per 100 000 women per year.[1 2] Estimates for Uganda show that cervical cancer led to 2300 deaths and 4000 new cases in 2012.[3] Majority of these deaths are preventable through human papilloma virus vaccination for young girls and screening for precancerous lesions for women at risk. In developed countries, where

### Strengths and limitations of this study

► This study involved focus group discussions with women and key informant interviews with health workers and administrators that enabled triangulation of responses thus increasing validity of the findings.
► Data collection took place in natural settings providing a conducive environment for women to openly express themselves and share experiences.
► The study was carried out in two majorly rural districts and results may not be generalisable to different settings.
► Study findings have the potential to influence design of future cervical cancer programmes and services.

prevention and control interventions have been implemented, reductions in detected cases and deaths have been registered over the past three decades.[4]

Cervical cancer screening rates in Uganda are very low despite the higher intention and willingness to screen. For example, a study conducted in central Uganda reported a screening rate of 7%,[5] while another in the eastern part of the country reported 4.8%.[6] Low cervical cancer screening rates have also been reported among health workers in Uganda.[7] Moreover, most cancers diagnosed in Uganda are already in advanced stage, when no remedial action can be taken.[8] Community-level factors such as: knowledge, attitudes and access to screening services can have implications for the successful implementation of screening programmes and other public health interventions.[6 9 10] Specifically, factors such as a feeling of embarrassment, perceived pain during examination, fatalism associated with a diagnosis, unsupportive husbands and lack of awareness about cervical cancer and available services can play a key role.[5 11–13]

Strategies such as community mobilisation and education, peer-to-peer engagement and organising health systems to track and follow-up women have been suggested to

mitigate barriers and optimise chances of success of screening programmes.[12 14] To be effective, such interventions require a deeper understanding of the social contextual factors influencing uptake of cervical cancer screening services. The few studies that have explored factors affecting uptake of cervical cancer screening in the east African region have mainly been quantitative and not had a deep exploration of such factors to fully inform future interventions. Therefore, the purpose of this study was to explore community knowledge, facilitators and barriers to cervical cancer screening among women in rural Uganda so as to generate data to inform interventions.

## METHODS

### Study area, design and population
This cross-sectional study was carried out in eastern Uganda in two majorly rural districts of Bugiri and Mayuge, which are approximately 150 kilometres from Kampala, the country's capital. These districts have a combined population of 856152 of whom 51.4% are females[15] and occupy approximately 10372 km$^2$. The major economic activities in the districts include: subsistence agriculture, fishing and operation of small businesses in trading centres. The study involved focus group discussions (FGDs) with women aged between 25 and 49 years who had lived in the study districts for a minimum of six (6) months. This age group was chosen because of their eligibility to access screening and higher risk of cervical cancer.[4] Key informant interviews (KIIs) were also carried out with members of the district health teams including the district health officers and health workers in both public and private health facilities at different levels such as nurses, medical officers, clinical officers and midwives.

### Sampling
We randomly selected five subcounties in each district (out of nine from Bugiri and seven from Mayuge) and purposively chose one village from each from which the FGD participants were identified. Villages were chosen to facilitate a wide geographical coverage of the study area as well as subpopulations of women in terms of rural-urban residence and socioeconomic status. One FGD was then conducted per selected village, making a total of 10 FGDs. Local leaders and community health workers in the selected villages guided the identification and purposive recruitment of participants fulfilling the inclusion criteria. The participants were then approached and invited to participate in the FGDs. Participants of the KIIs were purposively selected basing on their technical involvement in decision making relating to cervical cancer screening services provision. Eleven (11) KIIs were conducted in the two districts. These numbers of FGDs and KIIs enabled a theoretical saturation, when no new ideas were emerging.

### Data collection
Thematic guides for FGDs and KIIs, developed basing on previous literature and pretested among a similar study population, were used during data collection that took 2 weeks. The FGD guide explored knowledge and beliefs of women about cervical cancer, access to cervical cancer screening including barriers and facilitating factors and recommendations to improve screening uptake among women. However, the KII guide had questions on community knowledge and perceptions about cervical cancer, access to screening services, health system capacity to carry out cervical cancer screening and opinion on measures to increase utilisation of screening services. Both guides had several probes and prompts to guide the research assistants during data collection. The FGDs, which had an average of 12 participants, were conducted by two trained female research assistants with vast experience in conducting qualitative research and native speakers of *Lusoga*, the main local language used in the study area. No research assistants had any relationship with the study participants before the start of the study. The research assistants were supervised by a member of the research team throughout the data collection process for quality control. One research assistant moderated and facilitated the discussions, while the other assisted in taking notes and recording the interviews. FGDs were conducted in community gathering places identified by the research team in collaboration with local leaders. Sites for interviews were carefully selected to reduce interference from non-participants, and all study participants were encouraged to openly discuss their opinions. Local leaders arranged the venues, mobilised the participants and agreed with them a convenient time for the discussion. FGDs lasted on average 1 hour, excluding 10–15 min that research assistants spent administering consent and building rapport. For KIIs, after key informants were identified, appointments were made by phone, and interviews were scheduled at convenient times. The interviews were conducted by the researchers in places convenient to the informants mostly at their work place lasting an average of 45 min. All FGDs and KIIs were audio-recorded.

### Data management and analysis
The labelled recordings of the discussions and interviews were fully transcribed verbatim from the local language to English and proof read several times by the research assistants. The transcripts were then read several times by all research team members for familiarity with data and any emerging themes noted in addition to others identified in advance. All researchers, RN (environmental health scientist), TM (public health specialist), JK (anthropologist) and DM (public health specialist) have experience in designing and conducting qualitative research. Three of the researchers (RN, TM and DM) are males while JK is female. Two researchers (RN and TM) independently developed the code book for data analysis and described the coding tree, which were then reviewed and discussed with other researchers and any differences harmonised.

Data were coded after interpretation and ascribing meaning, and analysed using directed content analysis[16] with the help of Atlas.ti V.6.0.15 qualitative data management software. From the analysis, direct quotations from the FGD participants and KIs are presented in italics to highlight and support key findings. The consolidated criteria for reporting qualitative research (COREQ-32) checklist[17] guided the reporting of this study.

## Ethical considerations

The study protocol was reviewed and approved by the Higher Degrees Research and Ethics Committee at Makerere University School of Public Health and the study registered with the Uganda National Council for Science and Technology. Written informed consent was obtained from all participants after the moderator explained the study aims, benefits and potential risks. Participants' anonymity was maintained throughout the research process through use of numbers and confidential treatment of data.

## RESULTS

### Sociodemographic characteristics of participants

All participants approached to take part in the FGDs agreed to do so. Among the 119 FGD participants, the mean age was 34.4 years (SD±7.0) and most (50; 42.0%) were aged between 31 and 40 years, married 93 (78.1%) and had attained primary as highest level of education 58 (48.7%). Most participants were peasant farmers (69; 58.0%) and majority (82; 68.8%) had lived in the area for up to 20 years (table 1).

### Knowledge and beliefs about cervical cancer

This theme explored women's knowledge and awareness regarding causes, signs and symptoms, risk factors and screening methods for cervical cancer. In addition, it explored women's beliefs of being at risk of cervical cancer and whether and how the disease can be prevented and treated.

### Causes of cervical cancer

Almost all participants had heard about cervical cancer mainly from radios with the rest getting information from health workers at health facilities and a few from traditional health practitioners. However, knowledge about cervical cancer was poor, and several misconceptions existed among participants. In fact, many participants across the different FGDs consistently stated that cervical cancer resulted from use of contraceptives. This notion was majorly related to the side effects of some contraception methods including excessive bleeding and effects on menstruation cycles.

*I say contraception causes cervical cancer because when you start using certain options like swallowing pills, you find yourself bleeding so much meaning they affect you a lot, you bleed and bleed without getting treatment.* (FGD 7, participant 9, 40 years)

**Table 1** Sociodemographic characteristics of participants

| Characteristics | Frequency (n=119) | Per cent (%) |
|---|---|---|
| **District** | | |
| Bugiri | 50 | 42.0 |
| Mayuge | 69 | 58.0 |
| **Age (years)** | Mean=34.4, SD±7.0 | |
| 25–30 | 47 | 39.5 |
| 31–40 | 50 | 42.0 |
| 40–49 | 22 | 18.5 |
| **Marital status** | | |
| Married | 93 | 78.1 |
| Not married | 26 | 21.9 |
| **Education level** | | |
| None | 6 | 5.0 |
| Primary | 58 | 48.7 |
| Secondary | 51 | 42.9 |
| Tertiary/university | 4 | 3.4 |
| **Occupation** | | |
| Peasant farmer | 69 | 58.0 |
| Business | 22 | 18.5 |
| House wife | 16 | 13.4 |
| Others (health worker and teacher) | 12 | 10.1 |
| **Duration in area (years)** | Mean=16.2, SD±11.7 | |
| Less than 10 | 41 | 34.4 |
| Between 10 and 20 | 41 | 34.4 |
| More than 20 | 37 | 31.1 |

Some attributed cervical cancer to the foods they ate and its preparation including use of polythene bags to cover food when cooking while others highlighted other factors such as: having either many or fewer children, long use/non-use of sanitary pads, abortions, sharing hygiene facilities like bathrooms, and improper personal hygiene.

The key informants also confirmed the belief among women that cervical cancer was due to use of contraception methods.

*Many women state that they got cervical cancer due to use of contraceptives because it too is associated with bleeding. They forget that even before we started using contraceptives, cancer was in existence.* (Health worker at private facility, Mayuge)

Although none of the participants said that cervical cancer was associated with witchcraft, several KIs highlighted this as a perception that existed among women in the community. This they said also contributed to women seeking screening late as they could first go to the

traditional health practitioners before interacting with the formal health system.

*The community is ignorant and associate cervical cancer with witchcraft. They say that 'a co-wife or someone else bewitched me so that is why I am bleeding. (Health worker at government health facility, Bugiri)*

### Signs and symptoms of cervical cancer

Knowledge of signs and symptoms of cervical cancer was very poor and participants continuously stressed that they needed to be educated more to understand how the disease manifests.

*People don't even know the symptoms of cervical cancer for instance as one would know that a high fever and/or vomiting is associated with malaria but for cervical cancer we don't know the symptoms. (FGD 4, participant 11, 40 years)*

The few who knew some signs and symptoms of cervical cancer had either had a personal experience with the disease or with those affected by it. The major symptoms they listed were vaginal bleeding, backache and abdominal pain.

*My aunt began with bleeding abnormally and yet she had stopped menstruation. This persisted and she started seeking for treatment with the hope of becoming well. After some time, they diagnosed her with late stage cervical cancer and told her it would not be cured. (FGD 9, participant 8, 43 years)*

### Risk factors for cervical cancer

Knowledge about risk factors for cervical cancer was good as many participants noted that having: many sexual partners, sexually transmitted infections and family history of cervical cancer would increase one's risk. This was also noted by some KIs.

*I think having many sexual partners could lead to cervical cancer because the men also get various infections like gonorrhea and some are even HIV positive. Therefore having many sexual partners could bring cancer to you. (FGD 4, participant 13, 37 years)*

### Screening methods for cervical cancer

Knowledge about methods used for cervical cancer screening was poor with many participants stating that they had never heard of any method.

*We have never suffered from cervical cancer and neither have we screened for the disease. We thus don't know any methods used during screening. (FGD 10, participant 5, 45 years)*

However, a few participants described the methods they had experienced or heard of from their peers.

*The method that I know is they make you lay on your back and there is an equipment they insert in your private parts so that the doctor examining you can see the opening of the cervix well and whether it has wounds or not. (FGD 5, participant 2, 42 years)*

*I heard that the equipment they use helps in getting a sample from the cervix which they test to find out whether you have cancer or not. (FGD 2, participant 1, 30 years)*

### Perception of risk of cervical cancer

Almost all participants reported that they were at risk of getting cervical cancer giving several reasons ranging from their use of contraception methods to their lack of knowledge about signs and symptoms and preventive measures of the disease. They made reference to some risk factors of the disease such as cervical cancer being hereditary and possibility of acquiring it during sex, which they said put them at risk. Many participants also perceived that cervical cancer was a big problem in their communities though could not estimate its prevalence.

*I think we are at risk because we don't know how this cancer is prevented and what we should use to avoid getting it so this lack of knowledge puts us at risk. (FGD 2, participant 2, 32 years)*

### Prevention of cervical cancer

Most participants believed that cervical cancer could be prevented after understanding the causes of the disease, how it can be avoided and screening to know one's status.

*Preventing cervical cancer is possible after understanding its causes but before that, we still have a long way to go. (FGD 7, participant 6, 26 years)*

*I want to know that I don't have cervical cancer like I know that I don't have HIV/AIDs so that I can avoid it. (FGD 9, participant 10, 25 years)*

### Treatment of cervical cancer and outcome

Asking whether cervical cancer could be treated, most participants agreed that something can be done when the cancer is in its early stages though stated that when there are delays to access treatment, the cancer would not be cured. Other participants stated otherwise.

*I think cervical cancer is treatable; if you get to know early, they test and treat you. On the other hand, when you seek care while it is already out of hand, it might not be cured. (FGD 10, participant 10, 35 years)*

### Facilitators of cervical cancer screening

Among the study participants, only 5 (4.2%) had ever screened for cervical cancer and these were in only 3 of the 10 FGDs. This notwithstanding, most women were interested in receiving cervical cancer screening regularly and cited reasons such as wanting to: know their status, prevent cervical cancer and obtain treatment if they had the disease. The participants also stated that this willingness was shared by other women in the community. Study participants who had screened for the disease mostly had had its signs and symptoms. The other facilitators noted

were: family history of cervical cancer and awareness of the disease/screening service.

### Experiencing signs and symptoms of cervical cancer

The greatest facilitator of cervical cancer screening among the study participants was having experienced signs and symptoms of the disease that prompted them to visit health facilities for consultation. The manifestation of signs and symptoms also determined the number of times one went to the facility and some women pointed it out as an important determinant for accessing the screening service even in the future.

*I used to go for cervical cancer screening whenever I could over bleed. So I have so far screened thrice. (FGD 4, participant 1, 49 years)*

*The reason I went for screening is because I used to get abdominal pain during my menstruation. Although I did not get my results, they gave me some drugs which I took and became fine. I have never gone back ever since. (FGD 2, respondent 10, 35 years)*

### Family history of cervical cancer

A few participants reported family history as a facilitator for accessing cervical cancer screening among women.

*I recently buried my sister because of this cancer and before she died her cervix had been removed. Following this, my mother was also screened and she was told that she had cervical cancer. (FGD 2, participant 5, 38 years)*

### Awareness of the disease/screening service

Some women had screened for cervical cancer because they had accessed information mainly through radios and health facilities regarding the disease and understood the importance of screening. Additionally, others knew about the availability of free screening services at some health facilities.

*There is a time they announced that there was free screening for cervical cancer at the health facility and I went there although I did not follow up on my results. (FGD 2, participant 10, 35 years)*

### Barriers to cervical cancer screening

When participants were asked about barriers affecting their uptake of cervical cancer screening, several subthemes emerged: lack of knowledge about cervical cancer and screening, health system challenges, fear of test outcome and consequences and financial constraints. These were in agreement with barriers highlighted by the key informants.

### Lack of knowledge about cervical cancer and screening

Throughout the FGDs, lack of knowledge about cervical cancer and screening was continuously highlighted as a barrier to accessing the service. In addition, study participants stated that due to their lack of knowledge, it was sometimes hard for them to access screening services

without relating their symptoms with the disease. The key informants also re-echoed this stating that most women do not take the initiative to access screening without the signs and symptoms of the disease.

*We are constrained from accessing the service because by the time you get the cancer, you will not have known the symptoms early enough yet without having any symptoms, people are lazy to go for cervical cancer screening. (FGD 5, participant 7, 45 years)*

### Health system challenges

The major health system challenges from the FGDs were the lack of health facilities offering screening, lack of awareness of services availability, services being far away from the community and the mistreatment of women by health workers at health facilities.

*From the mobilizations that we do, some people would like to undergo the screening but there is no nearby health facility that is offering the service. Then in the health facilities, some health workers are also rude to women and so some give up on going there and rather go to traditional herbalists for help. (FGD 2, participant 8, 28 years)*

The key informants mainly highlighted inadequate human resources, lack of proper training to carry out screening and lack of screening materials at health facilities as barriers to cervical cancer screening.

*We have two health facilities that offer the service but these do not have medical officers and the available clinical officers do not have the capacity to screen for cervical cancer thus limiting screening to only one major hospital. (Health worker at government facility, Bugiri)*

### Fear of test, outcome and consequences

Women expressed different concerns about the testing methods used during cervical cancer screening from the way they are handled by health workers to who handles them. They said they were uncomfortable undressing themselves before health workers especially if they were male. Some of these fears were attributed to their bad previous experience or that of their colleagues.

*There are doctors who come here once in a while and they call us for screening but they handle you like you are going to give birth. They roughly insert an equipment inside and you feel pain. Even if women come and they hear it's the same method, very few will go there. (FGD 9, participant 3, 34 years)*

*…. the part that is affected by that cancer makes us uncomfortable when it comes to showing the health worker. This testing method should at least be improved. (FGD 5, participant 13, 40 years)*

The key informants also expressed these fears as barriers that they attributed to culture, male health workers, and significant age difference where older women are handled by younger health workers.

*You know cancers cut across all ages, a client of 40 years may come to a health worker of 20 years and would be like, 'you are my daughter, how can I undress myself to you?' on the other hand, the youth would say 'how can I expose myself to my mother?'. So the attitude towards age difference becomes a problem.* (Health worker at private facility, Mayuge)

Women also expressed fear of a positive diagnosis. Others noted that women could equally fear finding out their HIV status if this were to be provided during the visit. Participants also stated that testing for cervical cancer could lead to unfavourable consequences especially among women including being left by their male spouses if found positive or not being able to have more children.

*Women don't screen because they think that if their husbands know that they have cancer, they will look for other women who don't have it and leave them. Some men think that if a woman has cervical cancer she can spread it to them or even produce children with cancer. Men are also concerned about the resultant treatment expenses.* (FGD 2, participant 6, 40 years)

### Financial constraints

Many women highlighted that they were also constrained by lack of finances to cater for transport to visit screening centres, screening costs especially when accessing service from private providers and treatment costs if found to have the disease.

*Lack of money for transport prevents us from accessing the service; sometimes you don't know the place and you don't have anyone to help take you there and neither do you have the money to go with.* (FGD 10, participant 3, 37 years)

This was also noted by a KI who worked in a private health facility where there was a cost attached to accessing the service.

*I think people who come for cervical cancer screening will need money which they may not have. Although the service is free in government health facilities, sometimes the reagents are not there and so people don't want to go there. When they come to a private setting, there is a cost attached to the service which can affect utilization.* (Health worker at private facility, Mayuge)

### DISCUSSION

This study explored women's knowledge, facilitators and barriers to cervical cancer screening in rural Uganda. From the study, knowledge about cervical cancer causes, signs and symptoms, screening methods and prevention was poor. This is consistent with findings from another study carried out in Uganda.[10] Common misconceptions about cervical cancer included the use of contraceptives being perceived as a cause of the disease that has previously been reported and discussed by other studies.[10 18 19] The urgency for health education campaigns on cervical cancer among the population can therefore not be overemphasised. However, awareness of some cervical cancer risk factors was high and most participants perceived themselves to be at risk of the disease. Although some study participants justified this using held misconceptions such as their use of contraceptives, it is still positive for cervical cancer control programmes. Moreover, risk perception is a critical factor in promoting precautionary health behaviour and has been a determinant of cervical cancer screening in previous studies.[7 13]

Although among the FGD participants, only 4 in 100 had been screened for cervical cancer, most participants showed interest in accessing the service regularly. Such high willingness for cervical cancer screening has also been documented by previous research in Uganda[5 20] and elsewhere.[13 21] Facilitators for cervical cancer screening among study participants were: experiencing signs and symptoms of cervical cancer, having a family history of the disease and being aware of cervical cancer and screening services, similar to those reported by previous studies.[6 22] The belief among study participants that screening should be accessed after experiencing cervical cancer signs and symptoms is negative for the success of screening programmes. This further reinforces the need to increase awareness about cervical cancer among women. In fact, the preference for screening being asymptomatic women, education campaigns should encourage women aged between 30 and 49 years to screen for cervical cancer at least once in their lifetime as recommended by the WHO.[4] Since availability of services alone is not sufficient to facilitate screening, measures should be put in place to publicise such services to increase their uptake including through use of mass media such as radio and television. Community sensitisations and mobilisation for cervical cancer programmes can also be contributed to by community health workers, an important cadre in supporting health systems especially in developing countries.[23 24] Indeed, community health workers have significantly contributed to cervical cancer screening and prevention efforts previously.[25 26]

In this study, barriers to cervical cancer screening reported were: lack of knowledge about cervical cancer and screening; health system challenges; fear of test, outcome and consequences; and financial constraints. These barriers are similar to those documented by previous researches.[13 14 19 22 27] To facilitate uptake of cervical cancer screening, sensitisations should be carried out to increase awareness about the disease and importance of screening. Also critical is the need to increase access to cervical cancer screening services within communities to address health system challenges such as long distances to health facilities and transport costs. To achieve this in a developing country perspective, there is need to adopt a community outreach model of service delivery where screening services are extended to the community regularly while building capacity at lower health centres. Integration of cervical cancer screening services with others accessed by women such as antenatal care, family planning or postnatal care could also

increase screening uptake. In their assessment among policy makers and the community of integration of HIV and cervical cancer screening services, Kumakech *et al* highlighted several concerns.[28 29] These include: limited health system capacity, potential consequences of integration, prolonged waiting times at the health facility and tiredness among women and health workers. It is therefore important that such factors are also considered before any integration of cervical cancer screening with other services. In addition, since some women according to this study associated cervical cancer with use of contraceptives, such integration should be properly designed and implemented or else would run the risk of re-enforcing this myth as well as affecting uptake of both services.

The need for more female staff to carry out screening due to the embarrassment felt by some when attended to by male staff was also highlighted. This concern among women has been reported by many previous studies.[14 19 22 27] This gap can be bridged by building the capacity of available female health workers some of whom may be of lower cadres such as nurses and midwives through task shifting to carry out cervical cancer screening of women with support from other staff. It is also important to have a mix of both younger and older staff to cater for all clients. Additionally, the importance of having a well-functioning health system cannot be overstated in this context. In this, the health facilities should have adequate capacity including skilled and professional workforce and required supplies and logistics to provide quality screening services to women. These measures should have a positive impact on cervical cancer screening rates.

This study had strengths and limitations. First, it solicited views from women and key informants using FGDs and KIIs respectively that allowed for varied responses and enabled triangulation of findings thus increasing the validity of the study. Second, data collection took place in natural settings providing a conducive environment for women to openly express themselves and share experiences. However, the FGD environment could have influenced participants to give answers that they perceive to be more socially acceptable. Also, during FGDs, sometimes there is dominance by outspoken participants that was minimised by having experienced moderators who ensured that all participants are involved in the discussion. Lastly, although the study was carried out in two majorly rural districts and results not be generalisable to the whole country, it reports important information that could be instrumental in the design of future cervical cancer programmes and services.

## CONCLUSIONS

Although perceptions towards cervical cancer and screening were positive, knowledge of study participants on cervical cancer was poor highlighting an urgent need to prioritise sensitisation and provision of communities with adequate information about cervical cancer. To improve cervical cancer screening, effort should be focused on reducing identified barriers and enhancing facilitators through measures such as raising awareness about the disease, strengthening health systems capacity and using female health workers to carry out screening.

**Acknowledgements** The authors would like to thank the study participants for their time and contribution to this study. Appreciation is also extended to the research assistants, Jackline Mwendeze and Prossie Aliwebwa for the support they provided during data collection.

**Contributors** RN, TM, JK and DM contributed to study design and implementation, and data analysis. RN and TM drafted the manuscript. DM and JK critically reviewed the manuscript. All authors read and approved the final version.

**Competing interests** None declared.

**Ethics approval** Higher Degrees Research and Ethics Committee at Makerere University School of Public Health.

**Provenance and peer review** Not commissioned; externally peer reviewed.

**Data sharing statement** No additional data are available.

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
