## [Reviewer comments · BMJ Open]

ARTICLE DETAILS

TITLE (PROVISIONAL)	Knowledge, facilitators and barriers to cervical cancer screening among women in Uganda: a qualitative study
AUTHORS	Ndejjo, Rawlance; Mukama, Trasias; Kiguli, Juliet; Musoke, David

VERSION 1 - REVIEW

REVIEWER	Dian Baker PhD, APRN California State University Sacramento USA
REVIEW RETURNED	23-Feb-2017

GENERAL COMMENTS	This manuscript was evaluate using the 21 Standards for Reporting Qualitative Research (SRQR) (Academic Medicine, Vol. 89, No. 9 / September 2014) Background/ Introduction: Manuscript is well-organized and written. The background section provides sufficient logic that supports the importance of researching barriers and facilitators to cervical cancer screening in Uganda Methods: SRQR criteria are met for the methods section. Authors may want to include more information to explain their qualitative approach and guiding theories. The research design is clearly described and presented in enough detail that the process could be repeated by other researchers. Sampling, data collection, and data analysis are clear and meet qualitative research standards. Additional limitations for the study may include, that within a focus group setting, women may be unduly influenced by peers in the focus groups. Women may provide responses they anticipate that the researchers may want to hear. Results and Discussion: Conclusion were based on the findings from the data, quotes were included that supported the thematic analysis. Validity of results may be strengthen by sharing results with a sample participants to confirm the themes and recommendations. Other: The authors provided for ethics board review, reveled funding sources, and addressed authors' role in the study. Thank you for the opportunity to review this manuscript.
---

REVIEWER	DR FATIMA Z ISA MODIBBO NATIONAL HOSPITAL ABUJA NIGERIA
REVIEW RETURNED	10-Mar-2017

GENERAL COMMENTS	The study is interesting and well written. see comments below: 1.Abstract- settings: mention clearly what the geographical location
---

	is for the two districts in Uganda. 2. Methods- include description of sampling method chosen. How were participants contacted and recruited. Did they have any prior relationship with any of the researchers? 3. Methods- Include refusal rates with reasons for refusal 4. Methods- Important information of the researchers is missing. Include researcher credentials, and occupation at time of study. 5. line 43. mentions the use of an inclusion criteria. kindly outline what the inclusion criteria are? did you consider any past relationships with a cervical cancer patient? 6. Methods- State duration for the study. 7. Methods- Include information on whether pilot testing of the FGD guide was performed. 8. The supplementary reporting guide used in this study has not been mentioned. In the data analysis section include what reporting criteria was used for this study.
--	--

VERSION 1 – AUTHOR RESPONSE

Reviewer: 1

Reviewer Name
Dian Baker PhD, APRN

Institution and Country
California State University Sacramento
USA

Please state any competing interests or state 'None declared':
none declared

Please leave your comments for the authors below
This manuscript was evaluate using the 21 Standards for Reporting Qualitative Research (SRQR) (Academic Medicine, Vol. 89, No. 9 / September 2014)
Background/ Introduction: Manuscript is well-organized and written. The background section provides sufficient logic that supports the importance of researching barriers and facilitators to cervical cancer screening in Uganda

Dear Reviewer, thank you for this complement.

Methods: SRQR criteria are met for the methods section. Authors may want to include more information to explain their qualitative approach and guiding theories.
The authors feel that they have included sufficient information for this section including a reference for further reading. We are however happy to provide any more specific information if advised by the reviewer.

The research design is clearly described and presented in enough detail that the process could be repeated by other researchers. Sampling, data collection, and data analysis are clear and meet qualitative research standards.
Thank you.

Additional limitations for the study may include, that within a focus group setting, women may be unduly influenced by peers in the focus groups. Women may provide responses they anticipate that the researchers may want to hear.

Thank you for these suggestions which we have included in the manuscript under the limitations section (page 19) and read as below.

On the other hand, the FGD environment could have influenced participants to give answers that they perceive to be more socially acceptable. Also, during FGDs, sometimes there is dominance by outspoken participants which was minimized by having experienced moderators who ensured that all participants are involved in the discussion.

Results and Discussion: Conclusion were based on the findings from the data, quotes were included that supported the thematic analysis. Validity of results may be strengthened by sharing results with a sample participants to confirm the themes and recommendations.

We acknowledge this fact which we hope to consider for our future studies. Thank you.

Other: The authors provided for ethics board review, revealed funding sources, and addressed authors' role in the study.

Thank you for the opportunity to review this manuscript.

Reviewer: 2

Reviewer Name

DR FATIMA Z ISA MODIBBO

Institution and Country

NATIONAL HOSPITAL ABUJA NIGERIA

Please state any competing interests or state 'None declared':

NONE DECLARED

Please leave your comments for the authors below

The study is interesting and well written. see comments below:

1. Abstract- settings: mention clearly what the geographical location is for the two districts in Uganda. We have specified the geographical location for the two districts in the abstract as advised and now reads.

Discussions and interviews carried out in the community within two districts in Eastern Uganda. See page 2.

2 Methods- include description of sampling method chosen. How were participants contacted and recruited. Did they have any prior relationship with any of the researchers?

Study participants were purposively selected based on a set eligibility criteria with guidance from the local leaders and community health workers (Page 5). They did not have any relationships with the researchers and the Research Assistants.

3. Methods- Include refusal rates with reasons for refusal

We did not have any refusals and thus didn't find it necessary to include this in the manuscript.

Nevertheless, we have now included a statement about this in the results section which reads as follows:

All participants approached to take part in the FGDs agreed to do so. (See page 8)

4. Methods- Important information of the researchers is missing. Include researcher credentials, and occupation at time of study.

This has been included within the methods section as follows:

All researchers, RN (Environmental Health Scientist), TM (Public Health Specialist), JK (Anthropologist) and DM (Public Health Specialist) have experience in designing and conducting qualitative research. Three of the researchers (RN, TM and DM) are males while JK is female. (See pages 6 & 7).

5. line 43.mentions the use of an inclusion criteria. kindly outline what the inclusion criteria are? did you consider any past relationships with a cervical cancer patient?

The inclusion criteria was mentioned in the Study area, design and population section (Page 5) and reads as follows:

'The study involved focus group discussions (FGDs) with women aged between 25 and 49 years who had lived in the study districts for a minimum of six (6) months.' Past relationships with a cervical cancer patient was not a prerequisite for participation in this study.

6. Methods-State duration for the study.

Being a cross sectional study, data for this study was collected within a period of 2 weeks. We have included this in the write up in the data collection section. (See page 6)

7. Methods-Include information on whether pilot testing of the FGD guide was performed.

The FGD guide was pilot tested in a similar community before the study as already indicated in the data collection section on page 6.

8. The supplementary reporting guide used in this study has not been mentioned. In the data analysis section include what reporting criteria was used for this study.

We used the COREQ guidelines in reporting for this study as now indicated on page 7. A filled COREQ form with page numbers of where relevant information can be found within the manuscript has been attached.